# Feasibility of a Home-Based Mirror Therapy Program in Children with Unilateral Spastic Cerebral Palsy

**DOI:** 10.3390/healthcare11121797

**Published:** 2023-06-19

**Authors:** Anna Ortega-Martínez, Rocío Palomo-Carrión, Carlos Varela-Ferro, Maria Caritat Bagur-Calafat

**Affiliations:** 1Physiotherapy Department, Universitat Internacional de Catalunya, 08017 Barcelona, Spain; aortegam@uic.es (A.O.-M.); cbagur@uic.es (M.C.B.-C.); 2Physiotherapy Department, Fundació Aspace Catalunya, 08038 Barcelona, Spain; 3Physiotherapy Department, Universidad de Castilla-La Mancha, 45071 Toledo, Spain

**Keywords:** unilateral spastic cerebral palsy, mirror therapy, feasibility, home-based therapy, intensive therapy

## Abstract

Children with Unilateral Spastic Cerebral Palsy (US CP) have motor and somatosensory impairments that affect one side of their body, impacting upper limb functioning. These impairments contribute negatively to children’s bimanual performance and quality of life. Intensive home-based therapies have been developed and have demonstrated their feasibility for children with US CP and their parents, especially when therapies are designed with the proper coaching of families. Mirror Therapy (MT) is being studied to become an approachable intensive and home-based therapy suitable for children with US CP. The aim of this study is to analyze the feasibility of a five-week home-based program of MT for children with US CP that includes coaching by the therapist. Six children aged 8–12 years old performed the therapy for five days per week, 30 min per day. A minimum of 80% of compliance was required. The feasibility included compliance evaluations, total dosage, perceived difficulty of the exercises, and losses of follow-ups. All children completed the therapy and were included in the analysis. The total accomplishment was 86.47 ± 7.67. The perceived difficulty of the exercises ranged from 2.37 to 4.51 out of 10. In conclusion, a home-based program of Mirror Therapy is a safe, cost-efficient, and feasible therapy for children with US CP when the therapist is involved as a coach during the entire program.

## 1. Introduction

Cerebral Palsy (CP) is defined as “a group of permanent disorders of the development of movement and posture, causing activity limitation, that are attributed to non-progressive disturbances that occurred in the developing fetal or infant brain” [1]. Depending on the location and the extent of the brain lesion, motor impairments affect different parts of the body [2].

Unilateral Spastic Cerebral Palsy (US CP) is the second most frequent type of Cerebral Palsy, representing approximately 30% of the total [2,3]. It is caused by a lesion that affects one brain hemisphere, conditioning functional impairments. Even though these impairments are mostly observed in the contralateral side of the body, affecting both the upper and the lower limbs, the ipsilateral side can also suffer some movement limitations [4,5].

The most affected upper limb (AUL) suffers from different impairments in the areas highlighted by the International Classification of Functioning, Disability, and Health: body functions and structures, activities, and participation [6].

Even though spasticity is the main characteristic, especially in flexor muscles, problems related to limitations in the range of movement and low muscle strength are also present. These structural impairments mostly affect the distal parts of the AUL [7]. The somatosensory function of the hand is also affected, but the exact extent of the problem remains unclear [7]. In different studies, the percentage of children with these impairments ranges from 50% to 86% [7,8,9,10].

Both motor and sensory impairments have been demonstrated to be partly responsible for the functional limitations of the AUL. The unimanual capacity of the AUL is determined by these functional limitations. Those may include poor grip strength and dexterity, decreased velocity, and others [9,11,12]. Kinematic abnormalities, including mirror movements and movement deviations, are usually observed, especially at the distal parts of the AUL [8,13,14,15]. Recent investigations suggest that a wide extent of the unimanual lack of movement control relapses in the presence of mirror movements, which may be related to the corticospinal tract reorganization [15,16].

A poorer unimanual capacity of the AUL is related to poorer bimanual performance of children with US CP. This relationship may explain the fact that children with US CP show low performance in the activities of daily living that require performance and coordination of both hands [17]. This finding is also correlated with the functional level in the Manual Ability Classification System (MACS) [7,13]. Usually, children with US CP are classified in the first three levels of MACS, mostly in level II [18].

Some therapies have been highly recommended to improve both unimanual capacity and bimanual performance in children with US CP, especially for those classified in MACS levels I and II. Nevertheless, although the combination of Constraint Induced Movement Therapy and bimanual training seems to be a suitable approach, different strategies and other approaches have also been used and have demonstrated their effectiveness [19,20,21,22].

Considering that intensity is an important predictor for the success of the treatment, it can be assumed that in the last two decades, the literature recommending intensive approaches for improving motor activities and function in children with CP has increased [21,22]. Intensity, meaning the duration of the intervention and the distribution of the total dosage of the targeted therapy, may play a key role in all these approaches. That fact is especially crucial when considering the best therapy and its suitability for responding to parents and children’s goals [23].

Another key factor when considering the best approach is the setting of the therapy. A natural environment offers the possibility to increase the intensity and repetition, especially when compared to interventions completely set in a clinical environment [20].

In general, family-centered care is correlated with improvements in the well-being of children with disabilities and their families, as it enhances the participation of parents in the interventions and focuses on real context approaches [24]. Moreover, therapies performed at home with the supervision of parents have been demonstrated to increase the enjoyment and positivity of families about their children, as they are capable of experiencing the improvements in a real context [25].

However, families can sometimes find it difficult to follow home-based programs. The most identified difficulties are related to the insights of the home-based approach, as fitting the therapy into the daily routines may become an issue. This difficulty increases when the designed program is too demanding or requires many hours of therapy. Other difficulties have been described, for example, when the therapies are extremely difficult for children and require working memory. In these cases, parents can feel overwhelmed trying to make their children follow them [25].

Some strategies may help to improve and facilitate the adaptation to follow a home-based intervention [25]. Among them, coaching models are recommended. In these models, the communication between therapists and families forms the basis of the clinical approach [24,25]. Tele-rehabilitation and video monitoring have recently been incorporated to improve the participation and achievement of goals in home-based therapies [26]. Thus, designing proper coaching strategies may be crucial to ensure high compliance in any home-based intervention. This design should also consider the type of coaching and its frequency to avoid burning out families. When parents act as either the providers or the supervisors for the therapies, compliance with the treatment is high [25].

Different home-based approaches for children with US CP have demonstrated their feasibility in terms of acceptability and safety [20,26,27,28,29].

Studies analyzing the effectiveness of Mirror Therapy (MT) have recently emerged. MT is a non-invasive therapy that consists of placing a mirror in the sagittal plane of a patient suffering from a unilateral condition affecting upper or lower limbs in such a manner that the view of the non-affected limb is superimposed on the image of the affected limb. This positioning results in an illusion of the movement of the affected limb [30].

In the last decade, research on MT for children with US CP has increased. The specific neurological mechanisms that underlie the effects of the mirror illusion in children with US CP remain unclear. Nevertheless, MT has been proven to be capable of increasing the excitability of the primary motor cortex of the brain hemisphere related to the AUL, especially when a bimanual task is performed [31,32].

MT has been studied as part of the therapeutic approach for children with US CP alone or combined with other interventions. Some of the benefits shown in different studies include the improvement of pinch and grasp strength [33,34], dexterity [35], upper limb function [33,36], gross motor skills [37], bimanual skills [36,38], and activity performance [38], but to a limited extent. The repercussion of MT on other aspects defined by ICF-CY as quality of life still remains uncertain. [34,38,39,40,41].

From the different studies about MT, discrepancies can be found in the application, such as intensity, dosage, and the setting of the therapy. Some authors have studied MT or mirror conditions in a clinical environment [35,37,38,40,41]. Nevertheless, it is a suitable therapy to be performed at home [33,36,39] as it is a cost-efficient, easy, and approachable therapy. Thus, although there is some evidence supporting the benefits of this therapeutic approach, more research is needed to compare its effectiveness to other therapies that are currently strongly recommended. Moreover, identifying the main requirements for its implementation and a protocolization of the intervention is essential in order to recommend the implementation of MT in therapeutic strategies for children with US CP [21,22,29,34,42,43].

This study aims to evaluate the feasibility and effectiveness of a five-week home-based Mirror Therapy (MT) program for children with Unilateral Spastic Cerebral Palsy. Specifically, we aim to determine the necessary compliance and total dosage of the MT home-based therapy, assess the perceived difficulty of the exercises, analyze potential learning effects, and gauge overall program accomplishment.

## 2. Materials and Methods

### 2.1. Study Design

This is a feasibility study and includes the feasibility analysis of a group of children performing a five-week home-based Mirror Therapy program. This analysis is prior to a full single-blinded Randomized Clinical Trial (RCT) (NCT05244083). The RCT has been approved by the Ethics Committee of FIDMAG Germanes Hospitalàries (PR-2021-18) and the Ethics Committee of the International University of Catalonia (FIS-2021-07).

This study has followed the recommendations of the CONSORT 2010 statement for randomized pilot and feasibility trials [44,45].

### 2.2. Participants

The participants included six children between 8 and 12 years old diagnosed with Unilateral Spastic Cerebral Palsy and classified in levels I and II in MACS. The exclusion criteria were (a) botulinum toxin injections and/or extracorporeal shock waves applied in the affected upper limb in less than three months prior to enrollment; (b) surgeries in the affected upper limb six months prior to enrollment; (c) performing any intensive therapy for the affected upper limb; (d) moderate and severe intellectual disabilities; (e) existing comorbidities affecting attention or behavior; (f) non-corrected visual impairments; and (g) non-controlled epilepsies.

Two more exclusion criteria were applied for the families: at least one of the parents should be capable of answering questionnaires in Spanish, and they should have a table and a chair at home to perform the therapy.

All the participants were recruited from Fundació Aspace Catalunya (Barcelona, Spain) between January and September 2022. Once they were identified as potential participants who met the eligibility criteria, the principal investigator (A.O.-M.) sent an email inviting them to participate. Those who showed interest in participating were contacted by phone to set the training day. Previously to the training, informed consent was obtained from the parents of the participants.

### 2.3. Sample Size

As this was a feasibility study, a formal sample size calculation was not required [44]. Nevertheless, we aimed to recruit 20% of the total sample size of the RCT (*n* = 22). Finally, a total of six children were included in this feasibility study, representing 27.27% of the RCT sample size.

### 2.4. Intervention

The intervention consisted of a home-based program of Mirror Therapy that was designed to be performed for five weeks, five days a week, 30 min a day. A 5-min rest could be done in the middle of the therapy. All exercises were designed to be performed by the children themselves, while parents were involved in the supervision and maintenance of the concentration of their children. The MT program included four bimanual exercises (forearm pronosupination, sponge squeezing, finger-by-finger modeling clay pressing, and clockwise and anti-clockwise wrist spins), following exercises recommended by other authors [33,35,38,39,46] (see Figure 1). When doing the therapy, children were asked to insert the affected upper limb in a Mirror Box [47] (available from: www.noigroup.com, accessed on 1 January 2022) that was placed at their sagittal plane. All the exercises had to be done bimanually and symmetrically, highlighting the intention to move the AUL over the quality of the movement of the hand.

In order to increase motivation, children were told to do at least two of the four exercises suggested each day, letting them choose which exercises they wanted to practice and their order. Families were not told when they should do the therapy to increase the adaptability for the intervention at home. A minimum of 80% compliance with the total dosage (600 out of the total 750 min) was required.

The material needed for the MT intervention was provided to all the families. It included a Mirror Box [47], two cubes of modelling clay, and two sponges. The total cost of the material was lower than 70 € per child. Thus, the study created no cost for the families.

### 2.5. Follow-Up and Monitoring

Regarding the follow-up to the home-based intervention, a mobile health application was used. Each family was logged into the app, where they could find a video of each of the exercises. Every day, families were told to mark which exercises they performed, the exact amount of time they spent doing the exercises, and the child’s perceived difficulty performing the exercises. Nevertheless, in order to avoid difficulties with the usage of the app, a registration form was also given to all the families, allowing them to write down the same information that was requested in the app. A printed version of the explanation of the exercises was also given, replacing the videos in the app.

Moreover, the principal investigator (A.O.-M.) set up a weekly video call with each family to encourage and help them. When the video call could not be made, families were asked to send videos to the principal investigator doing the therapy.

A training day was scheduled before the beginning of the intervention in which both the child and the family were instructed by the principal investigator about the procedure of the intervention and the functioning of the app.

### 2.6. Variables and Measurement Instruments

During the entire duration of the study, three evaluations were established: at baseline, at the end of the intervention, and at the one-month follow-up. In these, the bimanual performance, the somatosensory function of the affected upper limb, and quality of life were assessed.

First, bimanual performance was assessed with the Children’s Hand-use Experience Questionnaire 2.0 (CHEQ 2.0) [48,49,50]. The CHEQ 2.0 is a valid and reliable questionnaire designed to assess the quality of the bimanual performance of children with Unilateral Cerebral Palsy from 6 to 18 years old. The Spanish version was used for this study available online (www.cheq.se, accessed on 1 January 2022). It analyzes grasp efficacy, the time taken, and the discomfort that children and/or parents perceive when performing 27 different daily activities, rating on a four-category scale. Then, the score was transferred to a 0–100 scale. CHEQ 2.0 can be answered by children or by parents. Nevertheless, the recommendation for children under 12 years old is for it to be answered by children helped by parents or only by parents as a proxy [49,50]. In this study, parents were the responders to the CHEQ 2.0.

Second, the somatosensory function assessment was performed according to six tests described and recommended by Auld et al. [51,52]. The assessment was done by a physiotherapist. A newly calibrated 20-item Semmes Weinstein Monofilaments (SWM) was used to test tactile registration [53], as well as for the single-point localization (SPL) and the double simultaneous (DS). The two-point discrimination tests, both static (s2PD) and moving (m2PD), were performed using a Disk-Criminator [54]. Of all these tests, SWM, s2PD, m2PD, and SPL are considered the most reliable in terms of detecting changes over time [51]. Finally, stereognosis was assessed with nine common objects [51,52]. The procedure of these tests can be found in the study of Auld et al. [51,52].

Third, quality of life was assessed with the Spanish version of Child and Parent Reports of the Pediatric Inventory of Quality of Life for Cerebral Palsy (PedsQL 3.0 ™), Cerebral Palsy module for children aged 8–12 [55]. PedsQL™ are valid and reliable questionnaires to assess the quality of life of children with Cerebral Palsy. For the range of age to which this study was addressed, the tool offers two questionnaires to be answered by the children and by the parents, respectively. Both are five-point Likert scale questionnaires that include 35 items regarding the areas of daily activities, school activities, movement and balance, pain and hurt, fatigue, eating activities, and speech and communication. After completing the questionnaires, the obtained score is transferred to a 0–100 point scale [55].

Due to the structure of the evaluations, the evaluator could only be blinded for the somatosensory function assessment. C.V-F acted as the blinded evaluator.

Apart from these assessments, age, sex, MACS level, and impaired side were recorded for all participants at the baseline evaluation.

### 2.7. Feasibility Evaluation

To evaluate the feasibility of the MT program, data regarding compliance, total dosage, perceived difficulty of the exercises, and accomplishment of the follow-up and evaluations were recorded. The appearance of side effects or contraindications was also recorded.

Regarding the recording of compliance with the intervention, as previously stated, the application allowed families and children to use a chronometer to quantify the total daily amount of time for the duration of the intervention. Therefore, the recording of the completed time was performed automatically. In addition, the application also allowed reporting of the exercises performed each day, as well as their perceived difficulty, through a very simple visual scale. It was rated from 0 (no perceived difficulty) to 10 (extremely difficult).

Finally, the accomplishment of the follow-ups was recorded weekly by the principal investigator in an Excel form. The completed evaluations were recorded using the same strategy.

Regarding the side effects or contraindications, all families were asked to document any problem detected during the entire intervention.

### 2.8. Statistical Analysis

For the analysis of the feasibility outcomes, descriptive statistics were used to show general characteristics, with means (standard deviation) for quantitative data and percentages for qualitative data.

Statistical analysis was only performed for the perceived difficulty outcome. In this, the Kolmogorov-Smirnov test was used to test normality. The Mann-Whitney U test was used to analyze intra-group effect differences. The significance level was set at α = 0.05. All analyses were performed by using the v.29 SPSS software package.

Due to the nature of this feasibility study, it was decided not to perform an effectiveness analysis on the bimanual performance, somatosensory function, and quality of life data. The same reasoning was used to decide not to perform statistical analysis regarding other relationships between age, sex, MACS level, impaired side, and feasibility outcomes [44].

## 3. Results

### 3.1. Participants

Six participants were included in this feasibility study. The mean age was 10.37 ± 2.05. Table 1 shows the main characteristics of the whole group.

### 3.2. Compliance and Total Dosage

All the participants acquired the minimum compliance required (80% of the total therapy), ranging from 80.0% to 96.0%. The mean percentage of compliance was 86.47 ± 7.67. When considering the total minutes of therapy performed, the mean was 648.55 ± 57.55. Table 2 shows the total minutes of MT performed by each child, and Figure 1 shows individual compliance.

### 3.3. Perceived Difficulty of the Exercises

The data extraction of the perceived difficulty of the exercises was performed considering *n* = 5 because one child participant did not complete this recording. This child was given a paper form registration for recording the therapy, as the family had problems with using the app. She did not record the difficulty due to the fact that she did not have the visual scale.

All the children rated the four included exercises in the MT program (forearm pronosupination, sponge squeezing, finger-by-finger modeling clay pressing, and clockwise and anti-clockwise wrist spins) below 8 when scoring their perceived difficulty. The forearm pronosupination and the sponge squeezing exercises were considered the easiest, with a weekly individual mean ranging from 2.37 to 3.17 and from 2.56 to 3.19, respectively.

No differences in the perceived difficulty were shown regarding age, sex, or MACS level of the participants (*p* > 0.05).

Table 3 shows the weekly mean (standard deviation) of the perceived difficulty of all exercises. No statistical differences were shown when comparing the evolution of the perceived difficulty of any exercise. These results may be affected by the loss of the complete registration of one child.

### 3.4. Losses of Follow-Ups and Evaluations

There was no loss of follow-ups in this study. All six children completed the entire therapy. Only one participant needed the paper form registration, as the family had difficulties with the app functioning. Although one child did not use the app, all six children recorded the daily therapy. The six families were available weekly either for a video call with A.O-M or to send videos performing the therapy.

All evaluations were completed at baseline and at the end of the intervention. Only one child did not entirely complete the PedsQL 3.0 ™ at the one-month follow-up evaluation.

### 3.5. Adverse Events or Contraindications

No adverse events or contraindications were observed. Families reported some fatigue only in the last minutes of the therapy, but this cannot be considered as an adverse event.

## 4. Discussion

This study aimed to analyze the feasibility of a home-based program of Mirror Therapy designed for improving motor and somatosensory impairments, as well as the quality of life of children with Unilateral Spastic Cerebral Palsy. Concretely, this study analyzes the compliance, the total dosage, the children’s perceived difficulty with the included exercises, and losses of follow-ups and evaluations.

Completing this feasibility study, previous to an RCT, has demonstrated that a five-week home-based program of MT is feasible for children with US CP and their families, as a high compliance rate and total dosages have been shown. Moreover, it has been stated that the included exercises were not considered to be very difficult, nor were they too easy. Last, this home-based MT program proved its feasibility, as there were no follow-ups were lost.

Other authors have previously studied MT in the same population. Bruchez et al. [39], designed a home-based protocol that included seven exercises comprising symmetrical movements of both distal and proximal parts of the upper limbs (finger-by-finger modeling clay pressing, thumb-index pinch-extension, palmar squeezing, wrist rotations, pronation and supination of the forearms, shoulder antepulsion and retropulsion, and shoulder abduction and adduction). Narimani et al. [35] designed a six-week clinical-based program of MT that included flexion and extension of the fingers and wrists, supination and pronation, and functional movements. The pilot study of Gygax et al. [33] described an MT intervention of three weeks that included three bilateral exercises (two regarding thumb-finger pinch and one of forearm pronosupination). Kara et al. [38], proposed a combination of MT and upper limb strength and power exercises. The MT was performed by doing four exercises (two of thumb-finger pinch, one of grasping a ball, and one of forearm pronosupination) for 30 min, 3 days a week, for 12 weeks. Auld et al. [40] designed an MT intervention that combined motor and tactile exercises, during 1.5 h in two sessions. Farzamfar et al. [37] included several exercises that involved the whole upper limb. Palomo-Carrión et al. [36] designed a four-week intervention with MT combined with Action-Observation Therapy. In this last study, MT was performed for 15 min per day and included six exercises.

In our study, the intervention was designed to comprise 750 min of MT distributed for 30 min daily for five weeks. This routine is consistent with other studies implementing MT where the designed total dosage ranged between 94 and 1080 min [33,35,36,37,38,39,40]. The distribution of the therapy is not inconsistent among studies. While other studies distributed the therapy between 2 days and 12 weeks, our study designed a five-week therapy. Moreover, the duration of the sessions varied between 15 min and 90 min [33,35,36,37,38,39,40]. Nevertheless, a duration of 30 min for each session, as our study proposed, was the most designed intervention [35,37,38]. A systematic review showed that different home-based programs for children with US CP lasted between two weeks and six months, with an intensity between 70 min and 56 h a week [25]. In these terms, our study was designed like these recommendations with regard to the total dosage and the distribution of the therapy. A five-week duration was the minimum duration for an intensive program for Myrhaug et al. [20]. This study also accomplished that specification, as all children completed the five-week program. For Beckers et al. [56], a duration of 12 weeks was considered too difficult for families and children.

Compliance of more than 85% was shown in this study. This result agrees with other studies analyzing the feasibility of different home-based programs, where compliance from 56% to 99% was shown [25,36,57,58]. The compliance in an MT intervention was only reported by Palomo-Carrión et al. [36], where 96% of the total dosage of the intervention was completed by all participants.

Our high compliance rates could be explained by many factors. First of all, the fact that parents acted as providers of the therapy could engage families to participate and continue until the end of the intervention. It has been stated that home-based therapies provided by parents are the most common home-based approach, as it increases confidence and satisfaction of families with the therapies without giving a therapist role to parents [24,25,56,59]. The family-provider approach has been studied and recommended as a key aspect of treating children with CP. Nevertheless, there is a lack of consensus in some terms regarding the relationship between the therapist and the family [24]. Although having coaching parents who act as providers of the therapy seems to be an important factor that may influence participation and compliance, most of the studies do not provide or report it specifically. Considering different studies that showed results of a home-based MT intervention, only those by Bruchez et al. [39] and Palomo-Carrión et al. [36] were provided by parents and coached by the therapist. Bruchez et al. [39] gave a DVD with instructions and the training regimen to the families, as well as the contact of the coordinator. In the same way, the families of the study by Palomo et al. [36] received a weekly online follow-up. Gygax et al. [33] also proposed a home-based intervention, but no coaching was described. Finally, the study from Auld et al. [40] varied from others, as the home-based intervention was provided by the therapist. None of these studies reported a rate of compliance, although we could expect high compliance in the study of Auld et al. [40], as it was therapist-delivered. Other studies with different home-based interventions used other ways to coach families. Some of them used webcam-monitoring during the performance of the intervention [60], while others maintained weekly phone calls and home visits [27,28].

In order to give complete coaching, an online follow-up with an app (that contained specific videos of each of the exercises), a weekly video call, and/or sending videos were required in this study. This coaching approach is similar to that used by Beckers et al. [56], where parents were asked to record the daily intervention and to send a video every week. Moreover, they utilized more strategies to resolve issues during the implementation of the home-based program, including home visits. Telemedicine has experienced an exponential increase since the COVID-19 pandemic. It has demonstrated that it can be an opportunity for reducing travels to clinics, and maintaining continuity and accessibility of treatments for children with CP [61]. The systematic review from Beckers et al. [25] described these strategies and others to coach families involved in home-based programs. Finding a balance between sufficient and excessive coaching for families involved in home-based therapy seems to be crucial. Parents sometimes reported the coaching received to be useful and valuable for them during the entire intervention, but having too much follow-up (e.g., too many phone calls or home visits) can put extra pressure on families [56].

When designing our therapeutic approach, we tried to include all these main ingredients to ensure good coaching during the five-week program, as we included the direct coaching of the therapist with online video calls and the video instructions of the therapy in the app. Moreover, we instructed the families to perform the intervention when they considered that it best fit their daily routine. All these factors may have been crucial for the great compliance and total dosage rates.

Other strategies for improving the preparation and confidence of the families included training sessions. In this study, a training session was set with every family in order to explain the therapy and the functioning of the app. Other authors have also implemented this strategy [57]. Apart from the training day and the video calls, all families could perform the entire intervention without reporting difficulties or the need for extra coaching.

Another remarkable strategy that we believe increased compliance and motivation of children was that we let them choose which exercises they wanted to practice every day. Other studies, like those by Bruchez et al. [39], Narimani et al. [35], and Kara et al. [38], described concrete protocols for intervention, where the number of repetitions and the order of the exercises were determined by the therapist or the coordinator. A different approach was utilized by Auld et al. [40] where motor and tactile exercises were included. They distributed a total amount of time for each stimulation but did not describe a specific order for the motor exercises. Other authors did not describe the distribution of the exercises during the intervention [33,37,46]. Engaging and motivating children seems to be a key factor when considering the benefits and the achievement of outcomes in family-centered approaches; it also may be increased when both children and parents are active subjects in the decision-making process of the intervention [24,56]. To our knowledge, this is the first MT program that allows children to choose the exercises performed in order to increase motivation. For us, this may be one of the reasons for having obtained no losses of follow-ups during the entire intervention, as well as at the last evaluation.

In this study, children were required to score the perceived difficulty of the exercises performed daily. Even though one child did not complete the difficulty assessment, we found that none of the exercises were considered very difficult, as all children scored them below 8. Gygax et al. [33] reported that half of the children reported difficulty maintaining concentration during the entire intervention but did not specifically find a correlation with a difficulty with the exercises. Palomo-Carrión et al. [36] reported that one family had trouble with the intervention. In our study, we decided not to increase the difficulty of the exercises, as MT itself can sometimes become quite difficult to perform or require children to focus on the performing. Other authors also decided not to increase the difficulty nor to change the exercises during the entire intervention [33,35,37,39,40,46]. Contrarily, Kara, et al. [38] reported an increase in the number of repetitions of the exercises and the level of difficulty when children were more capable of performing the exercises. Our results suggest that the difficulty of the exercises does not necessarily have to be increased, as we found no statistical differences in children’s perceived difficulty. Thus, children performing MT find the same difficulty in the exercises during the entire intervention, meaning that the exercises do not grow easier. As far as we are concerned, this is the first study of MT in children with US CP that includes the assessment of the perceived difficulty of the exercises. Our results in this area may also explain the fact that children did not fail to accomplish the entire intervention, as they did not find it too difficult or frustrating.

In terms of the acceptability of the intervention, all families accomplished these requirements and completed the registration every day. Only one family had difficulties with the usage of the app, so they were given a paper form for registration and a paper copy of the exercises.

Moreover, all families could perform the entire intervention with the initial kit of material they were given during the training day. Thus, participation in the study cost the families nothing. The total cost of the intervention was lower than 70 euros per child, meaning that the MT intervention was cost-efficient. A recent study in the Spanish population showed that children with CP attend around 22 consultations per year at the health services, with 14.6% of those in rehabilitation care [62]. Moreover, it is important to consider that the individual average cost of home-based care in children with CP is considerably lower than center-based care [63]. Family-centered care is still an approach that needs to be implemented more in healthcare services and requires further investment from different institutions. The low cost of the implementation of these therapies, considering the reduction of total costs and displacements, could be important arguments in favor of implementing them for both the healthcare providers and accessibility for all families [24,64].

None of the families reported any adverse events. During the video calls, some families only reported some fatigue in children when performing the therapy. Other studies also found no adverse events in MT or home-based programs for children with US CP [28,36,38].

All children included in this study completed the entire intervention. Thus, we obtained no losses of follow-ups. Nevertheless, one child did not fully complete one questionnaire for the last evaluation. Therefore, it was excluded. In other studies of MT, losses of follow-ups were related to failures in the execution of the intervention or evaluations [36,39], lack of cooperation of the children, or difficulties with transportation when the therapy was not set at home [38]. Our results align with those of other studies that did not report losses of follow-ups [33,35,37,40]. However, only one of the last studies was set at home, with parents delivering the intervention [33]. Comparing our results with other studies with the same population, other difficulties were described regarding the personal issues of the families [56].

This study has some limitations. First, a limitation has been identified regarding the nature of the MT. This therapy needs children to concentrate and focus on the mirror. Gygax et al. [33] reported this as an issue for their pilot study. However, we planned different strategies to enhance engagement, including the mobile health application following. As the mean compliance was high and all the participants acquired the minimum required, we could assume that this issue was not sufficiently remarkable to withdraw the study.

Another limitation has been identified regarding the recording of the difficulty of the exercises, as one child did not complete it. Even though the app functioned easily, using technologies may be difficult for some families. Given that we consider the app useful for most families, a more complete and detailed training session could be performed to avoid difficulties in this area.

To our knowledge, this is the first study to analyze the feasibility of a home-based MT program that includes specific and structured coaching for parents and evaluates the difficulty that children perceive of the therapy.

Proving the feasibility of this home-based program highlights the importance of implementing family-friendly therapies suitable to perform in a home-based context, facilitating its fitting into the daily routines of the families. The success of home-based therapies relapses partly in the support and coaching that families receive. Our results also highlight the importance of coaching parents during the entire intervention as they become providers.

Further studies comparing different coaching approaches for families and the suitability of this coaching in home-based therapies are needed to define an optimum follow-up for them.

Further research on MT is still needed in order to identify its effectiveness, as it has already demonstrated its safety, cost-efficiency, and feasibility when provided in a home setting with the coaching of the therapist.

Given the results of this study, we consider that the designed five-week home-based MT program has proved its feasibility. For this reason, we expect promising results in further RCT, where we will be able to analyze the effectiveness of this therapy in bimanual performance, somatosensory function, and quality of life of children with US CP.

## 5. Conclusions

A five-week home-based Mirror Therapy program is a feasible intervention for children with Unilateral Spastic Cerebral Palsy, demonstrating high compliance rates and no losses of follow-ups. Factors such as encouraging and coaching families during the entire home-based therapy may be crucial for maintaining and increasing motivation and obtaining high rates of compliance.

Designing comprehensive interventions that allow families to fit the therapies into their daily routines ease the follow-up of these type of approaches.

More research is needed to determine the effectiveness of Mirror Therapy in children with US CP to indicate its suitability in the therapeutic opportunities of these children.

## Figures and Tables

**Figure 1 healthcare-11-01797-f001:**
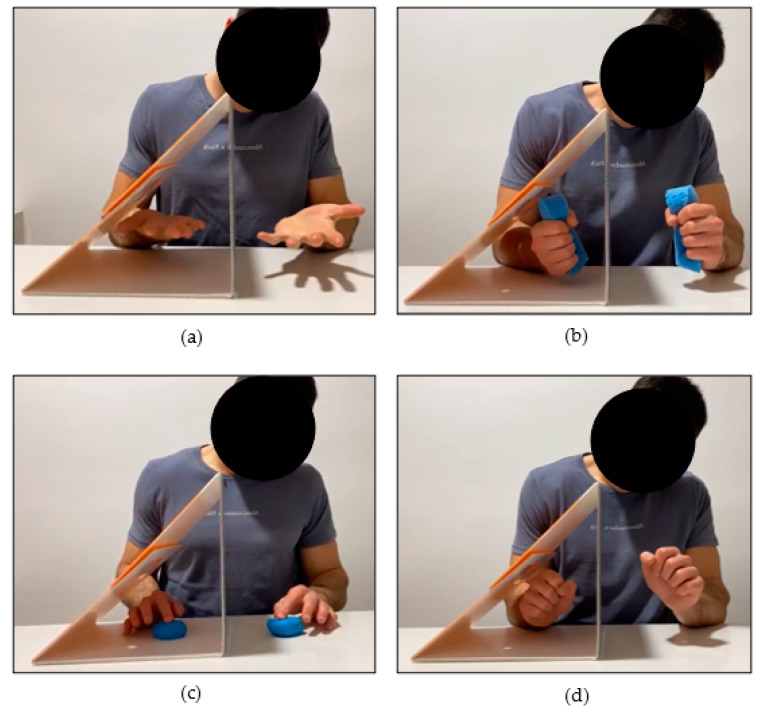
Mirror Therapy exercises. (**a**) Forearm pronosupination; (**b**) sponge squeezing; (**c**) finger-by-finger modeling clay pressing; (**d**) clockwise and anti-clockwise wrist spins.

**Table 1 healthcare-11-01797-t001:** Characteristics of the MT group.

Participants Characteristics	MT Group (*n* = 6)	
	n	%
**Sex**		
Male	2	33.3
Female	4	66.7
**MACS ^1^ level**		
I	3	50.0
II	3	50.0
**Affected Side**		
Left	1	16.7
Right	5	83.3

^1^ MACS: Manual Ability Classification System.

**Table 2 healthcare-11-01797-t002:** Compliance and total dosage of MT.

Participant	Total Minutes of MT
1	600.0
2	651.0
3	717.0
4	720.0
5	603.3
6	600.0

**Table 3 healthcare-11-01797-t003:** Weekly perceived difficulty.

Data (*n* = 5)	Forearm Pronosupination	Sponge Squeezing	Finger-by-Finger Pressing	Wrist Spins
Week 1	3.17 (2.51)	2.91 (1.44)	4.41 (2.49)	4.51 (1.81)
Week 2	2.38 (2.23)	3.19 (2.24)	4.46 (2.57)	4.32 (1.62)
Week 3	2.55 (2.32)	2.56 (2.38)	3.66 (2.59)	3.80 (1.41)
Week 4	2.37 (2.15)	2.82 (1.91)	3.23 (2.04)	4.07 (1.71)
Week 5	2.60 (2.71)	2.68 (1.90)	2.94 (2.04)	3.91 (1.98)
Differences between Week 1 and Week 5 (*p*-value)	0.345	0.715	0.225	0.068

## Data Availability

Not applicable.

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
