# Peer review of "Feasibility of a Home-Based Mirror Therapy Program in Children with Unilateral Spastic Cerebral Palsy"

_healthcare, 2023, doi:10.3390/healthcare11121797_

Round 1
Reviewer 1 Report
Dear Authors,
Thank you for submitting your manuscript entitled "Feasibility of a Mirror Therapy home-based program in children with Unilateral Spastic Cerebral Palsy" to the Healthcare journal. I appreciate the opportunity to review your work.
In my opinion, the manuscript presents several methodological errors that need to be addressed. Please find my comments and suggestions below:
Comment 1: The English text contains several errors, including typographical and grammatical mistakes. Please have a native English-speaking expert review the manuscript and attach a certificate from the expert confirming the review.
Comment 2: Reformulate and rewrite the abstract with the following sections: Background, Objective, Methods, Results, and Conclusions. Ensure that each section provides the necessary information and context for the reader.
Comment 3: Indicate the type of study in the title (e.g., "a feasibility study").
Comment 4: In the introduction section, provide more information on Cerebral Palsy (CP) and its different types. Offer a clearer definition of Unilateral Spastic Cerebral Palsy (US CP) to help readers less familiar with the subject better understand the context.
Comment 5: In the introduction section, provide more details about the specific methodologies and techniques used in the mentioned therapies, such as Constrained-Induced Movement Therapy, Hand-Arm Bimanual Intensive Training, and Goal-Directed Trainings.
Comment 6: In the introduction section, discuss and compare the advantages and disadvantages of the different therapeutic approaches mentioned. This will help readers understand what factors may contribute to the effectiveness of each approach based on patients' individual needs.
Comment 7: In the introduction section, define the effects that can be achieved through mirror therapy, discuss any previous evidence in similar populations, and elaborate on the mechanisms of action of Mirror Therapy and how it can improve symptoms and quality of life in children with US CP. Additionally, discuss the limitations of this therapy and how it compares to other available interventions.
Comment 8: In the introduction section, discuss how families adapt to home-based programs. Explain in greater depth the challenges families face when adapting to home-based therapy programs and the specific strategies that can facilitate adaptation, such as coaching models, tele-rehabilitation, and video monitoring.
Comment 9: We do not consider the keyword "viability" appropriate. Remember that keywords are specific terms or phrases that describe the content of a research article or document. We recommend using MeSH terms for keywords.
Comment 10: After reviewing the methodology section of your study, we suggest breaking down this section into subsections to make it easier for the reader to understand. Since your research is a feasibility study prior to a randomized clinical trial, it is crucial that the information is organized in a clear and coherent manner. We recommend dividing the methodology section into the following subsections:
Study Design: Briefly describe the purpose of the feasibility study and how it relates to the subsequent randomized clinical trial.
Participants: Detail the inclusion and exclusion criteria, as well as the participant selection process, including sample size and characteristics of the children and their families.
Intervention: Describe the home-based Mirror Therapy program, including duration, frequency, exercises, and the role of families in supervising the intervention.
Follow-up and monitoring: Explain the use of the mobile app for tracking participant progress and how weekly video calls and video recordings were conducted to ensure the proper implementation of therapy.
Variables and measurement instruments: Identify the variables that will be evaluated in the study and describe the measurement instruments used to assess these variables.
Sample size: Explain how the sample size for the study was determined and if any specific method was used to calculate it.
Statistical analysis: Describe the statistical methods that will be used to analyze the data collected in the study and how statistical significance of the results will be assessed.
Feasibility evaluation: Describe the criteria used to assess the feasibility of the home-based Mirror Therapy program, including compliance rate, perceived difficulty of exercises, and any other factors relevant to the success of the program. By dividing the methodology section into subsections, readers can easily follow the study's design and implementation.
Comment 11: Add a feasibility study-specific checklist, providing additional information for the reviewer.
Comment 12: After reviewing the feasibility study results section, I have found the following issues that require attention and significant modifications:
- In the results section, 3.1 participants only describe the average age because the rest of the data is not indicated as it is mentioned.
- You state in your objective that you analyzed the quality of life, but I don't see any relevant data to this variable? What measurement instrument did you use to evaluate this variable? Where is the reported data in reference to the quality of life?
- Table 4, we think, is redundant, and this information could be indicated in the text in a much more synthesized way in the form of short text. There is no need for a table to perform a check of evaluations.
- In the first sentence of the results section, it says "Six participants were included in this feasibility study." However, in Table 1, the number of participants appears as "MT group (n = 6)". This could confuse readers. To clarify, the sentence can be rewritten as: "Six participants were included in the MT group of this feasibility study."
- In the "Perceived difficulty of exercises" section, the text states that "data extraction of the perceived difficulty of exercises was carried out considering n = 5, as one child did not complete the record." It would be helpful to clarify why a child did not complete the record and whether this affected the overall study analysis.
- In Table 3, the "Weekly Perceived Difficulty" data is presented with decimal values like "3.17 (2.51)". However, it is unclear what the numbers in parentheses represent. You should clarify these values in the table or in the accompanying text.
- Lastly, it is essential to correct the text for grammatical errors or inconsistencies. For example, in the sentence "Thus, this last questionnaire was excluded from the record," it would be better to say "Thus, this last questionnaire was excluded from the analysis."
- There is a very poor data analysis, which does not correlate with the strong statement indicated in the conclusions.
- We suggest the researchers expand the results with the following analyses:
Subgroup analysis: Perform a subgroup analysis based on participant characteristics, such as gender, MACS level, and affected side, to investigate any significant differences in treatment adherence, perceived difficulty of exercises, and evaluation outcomes.
Correlation analysis: Assess the correlation between participant age and outcome measures (adherence, perceived difficulty of exercises, evaluation outcomes). This could help identify if age has an impact on treatment effectiveness or participant experience.
Temporal trend analysis: Examine whether there is any trend in the perceived difficulty of exercises over time (weeks 1 to 5) and if there are significant differences between the weeks. Additionally, analyze whether there is any relationship between the perceived difficulty of exercises and treatment adherence over time.
Analysis of the relationship between treatment adherence and evaluation outcomes: Investigate if there is any correlation between treatment adherence and improvements in evaluations. This could provide insight into the importance of treatment adherence in the program's effectiveness.
Analysis of the relationship between perceived difficulty of exercises and evaluation outcomes: Explore whether there is any correlation between the perceived difficulty of exercises and improvements in evaluations. This could provide insight into how the perceived difficulty of exercises affects treatment outcomes.
Comment 13: In the discussion section, it would be helpful to begin with a clear and concise summary of the study's main findings, highlighting the importance and contributions to the field of mirror therapy in children with Unilateral Spastic Cerebral Palsy.
Comment 14: In the discussion section, delve deeper into the comparison between the current study and previous literature, highlighting the unique aspects and novel findings of this study. This will help contextualize the results and demonstrate how the study contributes to existing knowledge in this area.
Comment 15: In the discussion section, discuss the potential clinical implications of the findings, including how the study results could contribute to the development of more effective interventions for children with Unilateral Spastic Cerebral Palsy. This will help demonstrate the relevance and potential impact of the study.
Comment 16: In the discussion section, address the study limitations in a more structured manner and discuss how these limitations may have affected the results. Additionally, suggest possible ways to overcome these limitations in future studies.
Comment 17: In the discussion section, delve deeper into the feasibility and acceptability of the intervention from the perspectives of the children, their families, and healthcare providers. This can include exploring factors that contributed to the high compliance rate and discussing strategies to further improve engagement and motivation.
Comment 18: In the discussion section, discuss the cost-effectiveness and practicality of implementing the mirror therapy intervention, considering factors such as resource allocation, accessibility, and the potential to scale up the intervention in various settings.
Comment 19: In the discussion section, discuss future directions for research in this area, including the need for larger randomized controlled trials, exploration of different training approaches, and the development of tailored interventions that address the specific needs of children with Unilateral Spastic Cerebral Palsy.
Comment 20: In the discussion section, reference 24 is too redundant, and we believe there is evidence that can replace this reference, as it is reflected throughout the entire discussion.
Comment 21: You should expand and reformulate the conclusions according to the study objective and the new analysis of the results, supported by existing evidence.
Comment 22: The references do not comply with the journal's guidelines. Ensure that you follow the journal's reference guidelines.
Comment 1: The English text contains several errors, including typographical and grammatical mistakes. Please have a native English-speaking expert review the manuscript and attach a certificate from the expert confirming the review.
Author Response
Thank you for your comments. We have reviewed the entire document in order to improve it, in the suggested way done by you and other reviewers. In red, we answer to your specific comments in this document. Also, you can find the modifications in the text highlighted in yellow
Dear Authors,
Thank you for submitting your manuscript entitled "Feasibility of a Mirror Therapy home-based program in children with Unilateral Spastic Cerebral Palsy" to the Healthcare journal. I appreciate the opportunity to review your work.
In my opinion, the manuscript presents several methodological errors that need to be addressed. Please find my comments and suggestions below:
Comment 1: The English text contains several errors, including typographical and grammatical mistakes. Please have a native English-speaking expert review the manuscript and attach a certificate from the expert confirming the review.
A native speaker has revised the full text, and corrections have been made when necessary. The certificate of the English review is attached to the submitted article.
Comment 2: Reformulate and rewrite the abstract with the following sections: Background, Objective, Methods, Results, and Conclusions. Ensure that each section provides the necessary information and context for the reader.
The presented abstract contemplates the suggested structure, but without headings, following the specific criteria defined in the template of Healthcare. Nevertheless, we have revised he information included to ensure that it contains all the key elements of the sections suggested.
Comment 3: Indicate the type of study in the title (e.g., "a feasibility study").
The title has been modified, including the suggestion. The final title is: “Feasibility of a Mirror Therapy home-based program in children with Unilateral Spastic Cerebral Palsy: a feasibility trial”. The word “trial” has been chosen, following the CONSORT recommendations for feasibility studies (see reference 40).
Comment 4: In the introduction section, provide more information on Cerebral Palsy (CP) and its different types. Offer a clearer definition of Unilateral Spastic Cerebral Palsy (US CP) to help readers less familiar with the subject better understand the context.
In lines 35-39 of the Introduction section, more information regarding US CP has been added.
Comment 5: In the introduction section, provide more details about the specific methodologies and techniques used in the mentioned therapies, such as Constrained-Induced Movement Therapy, Hand-Arm Bimanual Intensive Training, and Goal-Directed Trainings.
We have modified the paragraph of the introduction section that included this information, to clarify this observation. We considered that modifying the paragraph would help more the reader, instead of adding more information about other therapies that are not Mirror Therapy, as the main aim of this paper is to analyse the feasibility of this approach. The modifications can be revised in lines 61-65.
Comment 6: In the introduction section, discuss and compare the advantages and disadvantages of the different therapeutic approaches mentioned. This will help readers understand what factors may contribute to the effectiveness of each approach based on patients' individual needs.
Following the same reasoning of the previous comment, we have stated the discrepancies that nowadays exist, as more than one therapeutic approach may be suitable and effective, especially for children with levels I and II in MACS.
Comment 7: In the introduction section, define the effects that can be achieved through mirror therapy, discuss any previous evidence in similar populations, and elaborate on the mechanisms of action of Mirror Therapy and how it can improve symptoms and quality of life in children with US CP. Additionally, discuss the limitations of this therapy and how it compares to other available interventions.
We have revised the Mirror Therapy description, and we have included information regarding the neurological effects stated in different studies. Moreover, the limitations regarding MT have been described in a more extended way, in order to clarify the need of more research to compare it to other available interventions, and its effectiveness in the quality of life of children with US CP. All these modifications can be found in lines 111-131.
Comment 8: In the introduction section, discuss how families adapt to home-based programs. Explain in greater depth the challenges families face when adapting to home-based therapy programs and the specific strategies that can facilitate adaptation, such as coaching models, tele-rehabilitation, and video monitoring.
In lines 85-102, more information has been added to clarify the limitations that families face when performing a home-based intervention. In addition, we have deepened into the strategies that can be adopted to coach families involved in a home-based intervention.
Comment 9: We do not consider the keyword "viability" appropriate. Remember that keywords are specific terms or phrases that describe the content of a research article or document. We recommend using MeSH terms for keywords.
We have revised the entire document, but we have not found the use of the word “viability” neither in the keywords nor the entire document. We wonder whether this may be a confusion. Nevertheless, we have revised the entire document to ensure the usage of appropriate words.
Comment 10: After reviewing the methodology section of your study, we suggest breaking down this section into subsections to make it easier for the reader to understand. Since your research is a feasibility study prior to a randomized clinical trial, it is crucial that the information is organized in a clear and coherent manner. We recommend dividing the methodology section into the following subsections:
Study Design: Briefly describe the purpose of the feasibility study and how it relates to the subsequent randomized clinical trial.
Participants: Detail the inclusion and exclusion criteria, as well as the participant selection process, including sample size and characteristics of the children and their families.
Intervention: Describe the home-based Mirror Therapy program, including duration, frequency, exercises, and the role of families in supervising the intervention.
Follow-up and monitoring: Explain the use of the mobile app for tracking participant progress and how weekly video calls and video recordings were conducted to ensure the proper implementation of therapy.
Variables and measurement instruments: Identify the variables that will be evaluated in the study and describe the measurement instruments used to assess these variables.
Sample size: Explain how the sample size for the study was determined and if any specific method was used to calculate it.
Statistical analysis: Describe the statistical methods that will be used to analyze the data collected in the study and how statistical significance of the results will be assessed.
Feasibility evaluation: Describe the criteria used to assess the feasibility of the home-based Mirror Therapy program, including compliance rate, perceived difficulty of exercises, and any other factors relevant to the success of the program. By dividing the methodology section into subsections, readers can easily follow the study's design and implementation.
The subsections indicated have been added, and some modifications have been made in the text to improve the general comprehension of the methodology section.
Comment 11: Add a feasibility study-specific checklist, providing additional information for the reviewer.
We have followed the CONSORT 2010 statement for randomized pilot and feasibility trials. A specification has been added in 2.1 Study design section (lines 150-151). We have attached the guideline to this review.
Comment 12: After reviewing the feasibility study results section, I have found the following issues that require attention and significant modifications:
- In the results section, 3.1 participants only describe the average age because the rest of the data is not indicated as it is mentioned.
Apart from the age, a full description of the characteristics of the participants is included in Table 1, as it is mentioned in the section 3.1 Participants.
- You state in your objective that you analyzed the quality of life, but I don't see any relevant data to this variable? What measurement instrument did you use to evaluate this variable? Where is the reported data in reference to the quality of life?
The information regarding the measurements of the quality of life are included in 2.5 section, as quality of life is one of the variables of the RCT (lines 234-242). We believe that it is now clearer, as having improved this section including different sub-sections may clarify the presentation of this information.
Moreover, we decided not to show data regarding the effectivity of the therapy (bimanual performance, somatosensory function and quality of life), as we consider that this study should only reflect the feasibility of the intervention. This decision was made according to the CONSORT 2010 recommendations for feasibility trials. As this consideration may lead to a misunderstanding, we have included a general explanation in section 2.7 (lines 270-274).
Here, we attach the information found in the CONSORT guideline, that was used to make this decision: “A range of methods can be used to address the objectives in a pilot trial. These need not be statistical. Providing information about the methods used ensures that findings can be verified on the basis of the description of the analyses used. The primary focus is on methods for dealing with feasibility objectives. These methods are often based on descriptive statistics such as means and percentages but might also be narrative descriptions (example 1). Typically, any estimates of effect using participant outcomes as they are likely to be measured in the future definitive RCT would be reported as estimates with 95% confidence intervals without P values—because pilot trials are not powered for testing hypotheses about effectiveness.”
- Table 4, we think, is redundant, and this information could be indicated in the text in a much more synthesized way in the form of short text. There is no need for a table to perform a check of evaluations.
Table 4 has been removed.
- In the first sentence of the results section, it says "Six participants were included in this feasibility study." However, in Table 1, the number of participants appears as "MT group (n = 6)". This could confuse readers. To clarify, the sentence can be rewritten as: "Six participants were included in the MT group of this feasibility study."
Table 1 has been corrected in order to clarify the information.
- In the "Perceived difficulty of exercises" section, the text states that "data extraction of the perceived difficulty of exercises was carried out considering n = 5, as one child did not complete the record." It would be helpful to clarify why a child did not complete the record and whether this affected the overall study analysis.
This idea has been improved, adding the reason of the lack of completion of the registration of the perceived difficulty (lines 290-293). Moreover, line 304-305 has been improved to advise the reader to take these results with caution.
- In Table 3, the "Weekly Perceived Difficulty" data is presented with decimal values like "3.17 (2.51)". However, it is unclear what the numbers in parentheses represent. You should clarify these values in the table or in the accompanying text.
Line 302 has been improved, showing the interpretation of data included in table 3. Moreover, we have clarified the style of presentation of this data in section 2.7 (lines 263-265).
- Lastly, it is essential to correct the text for grammatical errors or inconsistencies. For example, in the sentence "Thus, this last questionnaire was excluded from the record," it would be better to say "Thus, this last questionnaire was excluded from the analysis."
The entire document has been revised to clarify some aspects. Concretely, the sentence indicated has been removed, because of other corrections.
- There is a very poor data analysis, which does not correlate with the strong statement indicated in the conclusions.
- We suggest the researchers expand the results with the following analyses:
Subgroup analysis: Perform a subgroup analysis based on participant characteristics, such as gender, MACS level, and affected side, to investigate any significant differences in treatment adherence, perceived difficulty of exercises, and evaluation outcomes.
Correlation analysis: Assess the correlation between participant age and outcome measures (adherence, perceived difficulty of exercises, evaluation outcomes). This could help identify if age has an impact on treatment effectiveness or participant experience.
Temporal trend analysis: Examine whether there is any trend in the perceived difficulty of exercises over time (weeks 1 to 5) and if there are significant differences between the weeks. Additionally, analyze whether there is any relationship between the perceived difficulty of exercises and treatment adherence over time.
Analysis of the relationship between treatment adherence and evaluation outcomes: Investigate if there is any correlation between treatment adherence and improvements in evaluations. This could provide insight into the importance of treatment adherence in the program's effectiveness.
Analysis of the relationship between perceived difficulty of exercises and evaluation outcomes: Explore whether there is any correlation between the perceived difficulty of exercises and improvements in evaluations. This could provide insight into how the perceived difficulty of exercises affects treatment outcomes.
The main aim of this study is to analyse the feasibility of the MT program. In order to clarify the specific objectives, and to respond better to them with the results showed, we have clarified them in the introduction section, as recommended in the CONSORT guidelines for feasibility trials (lines 132-142).
Only a descriptive analysis of data regarding feasibility outcomes has been performed, as we followed the recommendation of CONSORT. “A range of methods can be used to address the objectives in a pilot trial. These need not be statistical. Providing information about the methods used ensures that findings can be verified on the basis of the description of the analyses used. The primary focus is on methods for dealing with feasibility objectives. These methods are often based on descriptive statistics such as means and percentages but might also be narrative descriptions (example 1). Typically, any estimates of effect using participant outcomes as they are likely to be measured in the future definitive RCT would be reported as estimates with 95% confidence intervals without P values—because pilot trials are not powered for testing hypotheses about effectiveness.”
By following this statement, we will perform a much more detailed and concise statistical analysis after completing the entire Randomized Controlled Trial, as we consider that the data of the RCT will be more precise in order to identify such effects.
Comment 13: In the discussion section, it would be helpful to begin with a clear and concise summary of the study's main findings, highlighting the importance and contributions to the field of mirror therapy in children with Unilateral Spastic Cerebral Palsy.
A paragraph has been added, to resume the main findings of this study (lines 323-328).
Comment 14: In the discussion section, delve deeper into the comparison between the current study and previous literature, highlighting the unique aspects and novel findings of this study. This will help contextualize the results and demonstrate how the study contributes to existing knowledge in this area.
Our main objective was to demonstrate the feasibility of a MT program, designed considering the coaching of parents, and letting children choose the exercises performed. We have included this idea in lines 425-428, and lines 482-486. We expect the RCT results to contribute to increase the knowledge on the effectiveness of MT in children with US CP.
Comment 15: In the discussion section, discuss the potential clinical implications of the findings, including how the study results could contribute to the development of more effective interventions for children with Unilateral Spastic Cerebral Palsy. This will help demonstrate the relevance and potential impact of the study.
We have deepened in this argument, in lines 492-511.
Comment 16: In the discussion section, address the study limitations in a more structured manner and discuss how these limitations may have affected the results. Additionally, suggest possible ways to overcome these limitations in future studies.
The limitations section has been revised and completed with more information (lines 480-491).
Comment 17: In the discussion section, delve deeper into the feasibility and acceptability of the intervention from the perspectives of the children, their families, and healthcare providers. This can include exploring factors that contributed to the high compliance rate and discussing strategies to further improve engagement and motivation.
Some paragraphs have been included to answer to this point: lines 403-408, lines 425-428, lines 444-448, lines 462-465, lines 482-486.
Comment 18: In the discussion section, discuss the cost-effectiveness and practicality of implementing the mirror therapy intervention, considering factors such as resource allocation, accessibility, and the potential to scale up the intervention in various settings.
This argument is discussed in paragraph from line 455 to 465. The highlighted sentence in this paragraph has been added to complete the idea.
Comment 19: In the discussion section, discuss future directions for research in this area, including the need for larger randomized controlled trials, exploration of different training approaches, and the development of tailored interventions that address the specific needs of children with Unilateral Spastic Cerebral Palsy.
We consider that we could be able to respond better to this argument when we finalize the complete RCT. For this reason, we have added this idea in lines 507-511.
Comment 20: In the discussion section, reference 24 is too redundant, and we believe there is evidence that can replace this reference, as it is reflected throughout the entire discussion.
We decided to include this reference in the discussion section, mostly related to the comparison between our intervention design and other designs of MT programs, and the reported outcomes regarding the feasibility of different approaches. As it is a pilot study, we keep in mind that the results presented should be considered with caution, especially when we will analyse and discuss the outcome results of the entire RCT.
Comment 21: You should expand and reformulate the conclusions according to the study objective and the new analysis of the results, supported by existing evidence.
The conclusions section has been improved.
Comment 22: The references do not comply with the journal's guidelines. Ensure that you follow the journal's reference guidelines.
Reference list has been revised and some errors have been corrected.
Comments on the Quality of English Language
Comment 1: The English text contains several errors, including typographical and grammatical mistakes. Please have a native English-speaking expert review the manuscript and attach a certificate from the expert confirming the review.
An English native speaker has revised the entire document. The certificate is attached to the revision (please see the attachment).

Reviewer 2 Report
Studies of home-based program in neurological field need to be slightly reflected in Introduction.
Six participants seem to be not enough even if it is feasibility study.
Organization and contents seem to be original and creative.
However, this author reasons about results need to be reflected more in line of 284-286 in Discussion.
Reviewer 3 Report
The idea is interesting and of value for children with hemiplegic cerebral palsy or hemiparesis. Activities of daily life are reatly affected in unilateral hand function disturbance.
Aim : "The aim of this study is to analyze the feasibility of a home-based program of MT in children with US CP" The aim is clear, the conclusion differs somehow from the aim.
Methods: "This study is a feasibility study. It includes the feasibility analysis of a group of children performing a Mirror Therapy program. This analysis is prior to a single-blinded Randomized Clinical Trial (RCT) (NCT05244083). The RCT has been approved by the Ethics Committee of FIDMAG Germanes Hospitalàries (PR-2021-18) and the Ethics Committee of the International University of Catalonia (FIS-2021-07)." It is a preparatory or pilot before the main study.
In the methods the authors mentioned many assessment tools for hand function and somatosensory assessment, are they for feasibilty testing or for assessment of MT effect on hand function? Did the author calculate the sample size to be 6?
The methods need to be simpler and related to the feasibility of home MT program to be done.
Six patients can be used for feasibilty for the MT and the online questionnaire and video recording, but will not be feasible for statistical analysis of the results.
Line 170 in the methods "rating into a four-category scales. Then, punctuation is transformed in a 0-100 scale" Punctuation and register used frequently, may be I can not understand the meaning. Is punctuation stands for score and register for record ?
Results: need modifications in the tables. I did not find a figure as stated in the text.
Table 4 about complete evaluations is just a checklist for what is completed, does not add any significant results. The 6 questionnaires can be added as supplement.
The discussion needs to be rewritten according to the aim and after correction of results.
The conclusion should be related to the feasibility of performing home based MT
The study is feasibilty study, it is a prestudy for a registered work.
The authors used 6 children only. They used many scales which need rewritting and removing what is not needed.
They should be concentrating on the feasibility of home mirror therapy program.
Round 2
Reviewer 1 Report
While the manuscript has significantly improved in content and methodological structure, residual errors remain. These must be rectified before its final publication. Despite the progress, these lingering issues could potentially affect the research's credibility. As such, a detailed review is needed to ensure the manuscript's overall quality. The following outlines the necessary changes to be made:
1º The term "Feasibility" is indeed redundant in the current title of the manuscript. For a more concise and impactful title, I would suggest removing the redundancy and rephrasing it. An example of a revised title could be: "Home-Based Mirror Therapy Program in Children with Unilateral Spastic Cerebral Palsy: A Feasibility Trial". This new title maintains the key information while removing the redundancy and enhancing readability.
2º It’s noted that the manuscript includes several self-citations, such as reference 32, authored by Palomo-Carrion R. While self-citations can occasionally be appropriate, it's essential to balance them with citations from other relevant, high-quality works in the field. Over-reliance on self-citations can give the perception of a narrowed perspective.
Therefore, it's recommended that the self-citations, specifically the one mentioned, be replaced by references from other authoritative sources. This will not only ensure a broad and comprehensive view of the topic but will also provide a more robust theoretical foundation for your research. This is in line with maintaining the highest standards of scholarly writing and academic integrity.
3º In reference to the abstract. However, upon careful review, it has been observed that the second comment from the previous revision letter, concerning the structure and content of the abstract, appears to have been overlooked.
In the prior round of reviews, it was specifically suggested that you revise the abstract to align with the key sections of the manuscript. While it's understood that section headings are not typically included in an abstract, the content should nonetheless reflect the essential aspects of each section. This alignment is particularly crucial given the significant changes that have been made in your manuscript.
Unfortunately, it seems that the current abstract remains unchanged and does not effectively incorporate or reflect these revisions. As the abstract is a concise representation of your study, it is vital that it captures all recent amendments accurately, enabling readers to quickly grasp the essence of your research.
Hence, I kindly urge you to revisit and revise the abstract, ensuring it aligns with the content of the key sections - Introduction, Objectives, Methods, Results, and Conclusion. This revision should encapsulate all the significant updates that have been made in the manuscript to accurately reflect the current state of your study.
4º While the introduction provides a comprehensive overview of the topic, there are a few areas that could be clarified and condensed to improve readability and focus. Here are my comments and suggestions:
- Redundancy: Several statements appear repetitive or overly similar. For example, the discussion about home-based approaches seems to be mentioned repeatedly without adding much new information each time. Carefully revising these sections to reduce repetition would make the introduction more concise and impactful.
- Depth and Specificity: While the manuscript covers a broad range of information, it would benefit from adding depth to certain sections. For instance, when discussing the characteristics of Unilateral Spastic Cerebral Palsy (lines 32-36), you could include more specific details or recent findings about its manifestations or impacts on affected individuals. This would give the reader a clearer and more detailed understanding of the subject.
- Objective of Study: The aim of your study could be clarified and more directly stated. At the moment, the objective is spread out over several lines (125-135), which dilutes its impact. Consider condensing this into a single, concise statement. Here's an example based on your text:
"This study aims to evaluate the feasibility and effectiveness of a 5-week home-based Mirror Therapy (MT) program for children with Unilateral Spastic Cerebral Palsy. Specifically, we aim to determine the compliance and total dosage of the MT home-based therapy, assess the perceived difficulty of the exercises, analyze potential learning effects, and gauge overall program accomplishment."
Lastly, it is crucial to ensure that all elements of the study's objective are thoroughly analyzed in the manuscript. Each objective should have a corresponding section in your results and discussion. While the mention of a future RCT is useful, it should not be the main focus of this manuscript. Readers are interested in the findings of your current study, and I recommend focusing on presenting and discussing these findings in detail.
By addressing these points, your manuscript will offer a clearer, more concise, and more compelling introduction to your research.
5º In response to comment 11 from the previous review letter, it remains unaddressed. You have provided a clinical trial registration for an RCT which is entirely different from the study you are proposing. I believe a specific registration for the current study should be submitted.
6º In reference to comment 12, you mention that "we decided not to show data related to therapy effectiveness (bimanual performance, somatosensory function, and quality of life)". However, it's unclear why this information is omitted, as your study design for an RCT does include these variables. It's confusing that you neglect this data yet address other issues not covered in any registration. You refer to the CONSORT guidelines in terms of feasibility studies, but there's no registration for feasibility available. You provide a registration for an RCT. Either report the data specified in your design, or this practice could be seen as counterproductive and unethical.
7º A primary shortcoming in your manuscript appears to be the lack of comprehensive and pertinent data. While the mention of a prospective randomized controlled trial (RCT) has its merit, it should not constitute the central focus of this present manuscript. Readers are principally interested in the findings of your current study, and it is recommended that you prioritize presenting and discussing these results in greater depth.
Moreover, a feasibility analysis can offer significant statistical information and data. This can encompass recruitment and retention rates, protocol feasibility, intervention adherence, preliminary efficacy data, effect size and variance estimates, and safety data collection. It is strongly advised that you take these factors into consideration and incorporate this information into your manuscript.
I trust you will find these suggestions beneficial, and I encourage you to incorporate them into your manuscript to enhance the clarity and depth of your study's findings.
Minor editing of English language required.
